# A shape-based functional index for objective assessment of pediatric motor function

**Shashwat Kumar**[1☯], **Arafat Rahman**[1☯], **Robert Gutierrez**[1], **Sarah Livermon**[1], **Allison N. McCrady**[2], **Silvia Blemker**[2], **Rebecca Scharf**[3], **Anuj Srivastava**[4], **Laura E. Barnes**[1]*

**1** Systems and Information Engineering, University of Virginia, Charlottesville, Virginia, United States of America, **2** Biomedical Engineering, University of Virginia, Charlottesville, Virginia, United States of America, **3** Department of Pediatrics, University of Virginia School of Medicine, Charlottesville, Virginia, United States of America, **4** Department of Statistics, Florida State University, Tallahassee, Florida, United States of America

☯ These authors contributed equally to this work.

* lb3dp@virginia.edu

## Abstract

Clinical assessments for neuromuscular disorders, such as Spinal Muscular Atrophy (SMA) and Duchenne Muscular Dystrophy (DMD), continue to rely on subjective measures to monitor treatment response and disease progression. We introduce a novel method using wearable sensors to objectively assess motor function during daily activities in 19 patients with DMD, 9 with SMA, and 13 age-matched controls. Pediatric movement data is complex due to confounding factors such as limb length variations in growing children and variability in movement speed. Our approach uses Shape-based Principal Component Analysis to align movement trajectories and identify distinct kinematic patterns, including variations in motion speed and asymmetry. Both DMD and SMA cohorts have individuals with motor function on par with healthy controls. Notably, patients with SMA showed greater activation of the motion asymmetry pattern. We further combined projections on these principal components with partial least squares (PLS) to identify a covariation mode with a canonical correlation of $r = 0.78$ (95% CI: [0.34, 0.94]) with muscle fat infiltration, the Brooke score (a motor function score) and age-related degenerative changes, proposing a novel motor function index. This data-driven method has the potential to inform future home deployments with wearable devices, allowing better longitudinal tracking of treatment efficacy for children with neuromuscular disorders.

## Introduction

Emerging drugs, including gene and cell therapies, are rapidly developing as transformative treatments for rare and degenerative diseases. Duchenne Muscular Dystrophy (DMD), the most prevalent genetic cause of death in boys, and Spinal Muscular Atrophy (SMA), a leading genetic cause of infant mortality, have witnessed groundbreaking advancements with therapies such as anti-sense oligonucleotides and gene replacement therapies [1–3]. Despite

**Data availability statement:** An anonymized version of the dataset to replicate the results is

available at https://zenodo.org/records/16757630 and code is available at https://github.com/BarnesLab/UExtendPLOS.

**Funding:** Initials of the authors who received each award: Laura Barnes, Silvia Blemker, Rebecca J. Scharf Funder Name: Center for Engineering in Medicine, University of Virginia. Funder Name: Coulter Center for Translational Research at the University of Virginia. Website: https://engineering.virginia.edu/centers-institutes/coulter-center-translationalresearch No, the sponsors or funders did not play any role in the study design, data collection and analysis, decision to publish, or preparation of the manuscript.

**Competing interests:** The authors have declared that no competing interests exist.

these strides, the landscape of drug development remains hindered by significant challenges, primarily due to the difficulty in recruiting larger cohorts. This issue is further complicated by the subjective nature and imprecision of current trial outcome measures. These often rely on observational motor assessments, such as the Brooke Upper Extremity Scale, which measures arm function in patients with DMD [4], and the Children's Hospital of Philadelphia Infant Test of Neuromuscular Disorders (CHOP-Intend), which evaluates motor function in infants with SMA [5]. Both scales, along with other observational methods, may be susceptible to clinical bias, and may not capture subtle changes critical for evaluating treatment efficacy.

The emergence of wearable-based motion assessments presents a promising solution to these challenges. By embedding sensors into everyday activities, continuous, home-based monitoring becomes feasible, offering a holistic view of patient health beyond sporadic clinical visits [6–9]. This approach facilitates the collection of longitudinal data with greater ease and frequency, enabling more accurate tracking of disease progression and treatment effects over time [10–12]. In contrast to traditional methods that rely on intermittent clinical evaluations, wearable sensors allow for the seamless gathering of comprehensive movement data in a naturalistic setting, reducing the burden on patients and their families [13,14].

However, pediatric movement data is inherently complex, due to confounding factors such as limb length variations in growing children, variability in movement speed, and differing cognitive and developmental abilities. These issues can significantly alter movement trajectory representations, complicating the analysis and comparison of motion trajectories, especially in a young population where consistent movement speeds are difficult to achieve [15,16]. Robust methods for temporal alignment are essential for accurately comparing and analyzing trajectories to understand variables such as disease progression across various ages, phenotypes, and stages of the disease.

Moreover, many existing classifiers in digital medicine rely on black-box features [17–21], making it challenging for clinicians to trust their outputs [22,23]. In order to address these challenges, we utilize Shape-based Principal Component Analysis to simultaneously temporally align movement trajectories and quantify patient behavior in terms of interpretable shape-based phenotypes [24–26]. This method identifies and correlates specific movement patterns with clinical metrics such as muscle fat infiltration and motor function scores. By providing transparent and intuitive results, our approach has the potential to provide objective feedback on treatment progress compared to existing methods.

## Materials and methods

### Overview of the approach

Fig 1 presents a comprehensive workflow for analyzing Activities of Daily Living (ADL) using sensor-based data and various clinical measures, described in Table 1. Initially, raw sensor signals (X) are collected during ADL tasks. These signals are then aligned or registered using phase amplitude separation [27] and subjected to Shape-based Principal Component Analysis (PCA) in the shape space. The scores from this shape space are analyzed using Partial Least Squares (PLS) analysis to explore the covariation between the sensor signals (X) and multiple outcome measures (Y), including age, ultrasound measures (Cross Sectional Area, Average Echogenicity), dynamometer measures (Normalized Elbow Torque), and Brooke scores. The aim is to understand the relationships and potential predictive power of sensor data concerning these outcome measures, despite the absence of a gold standard for Y.

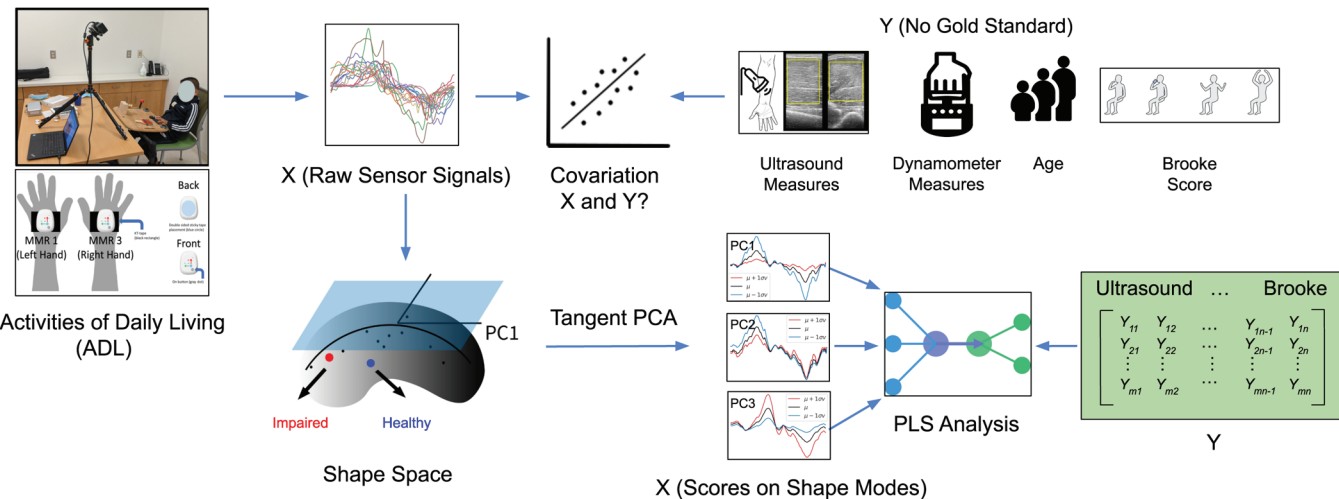

**Fig 1. Overview of the study and the proposed shape analysis pipeline.** Wearable sensors capture IMU signals from participants performing activities of daily living. This data is combined with shape analysis and external assessments to develop a canonical index of motor function.

**Table 1. Description of clinical measures against which we correlate our wearable features.**

| Clinical Measure | Description |
|---|---|
| **Brooke Score** | The Brooke Upper Extremity Scale is a 6-point ordinal scale used to classify upper limb function and track progression in neuromuscular disorders. A higher score indicates greater impairment [4]. |
| **Cross-Sectional Area (CSA, cm$^2$)** | Represents the anatomical size of a muscle. Larger CSA generally implies more muscle fibers and higher force-generating capacity [28]. |
| **Normalized Elbow Torque (NET, Nm/cm)** | Elbow torque normalized by forearm length to allow for comparison across individuals. |
| **Average Echogenicity (Avg_Echo, gsv)** | Echogenicity quantifies a muscle's ability to reflect ultrasound waves. In SMA, motor neuron degeneration leads to increased echogenicity due to fibrous and fatty tissue replacement [29]. In DMD, loss of dystrophin similarly results in muscle degradation and fat infiltration, raising echogenicity [30]. |

## Experimental protocol

This study, approved by the University of Virginia's Institutional Review Board for Health Sciences Research (protocol #HSR200178), recruited participants through the Pediatric Neuromuscular Clinic at the University of Virginia Children's Hospital [31]. Patients diagnosed with either SMA or DMD participated, along with age and sex-matched healthy controls (N = 13). The recruitment started on February 11, 2021 and ended on September 19, 2021. All adult participants and guardians of minor participants gave their written consent. All participants' demographic data are shown in Table 2. Participants wore MetaMotionR+ (MbientLab, San Francisco, CA, USA) sensors on both dominant and non-dominant hands, with accelerometer and gyroscope data collected at 200 Hz [32]. Activities of daily living (ADLs) including rotating a door knob, raising a cup, arm curl, door knocking, and moving a paddle were performed by the participants. The Brooke Upper Extremity Scale was employed to provide a standardized metric for comparison across all cohorts [4]. Following data processing, a subset of participants were excluded from subsequent analysis due to sensor malfunction (N = 2), young age and refusal to cooperate (N = 2), deceased (N = 1), participant withdrawal

**Table 2. Demographics of participants.**

| Cohort | Healthy | SMA | DMD |
|---|---|---|---|
| Participants (N) | 13 | 9 | 19 |
| Age Range | 2–35 | 2–19 | 4–35 |
| Mean Age ± SD (yrs) | 15.2 ± 10.6 | 7.4 ± 6.3 | 14.2 ± 9.4 |
| Sex (M/F) | 8/5 | 2/7 | 18/1 |
| Ambulatory (N) | 13 | 4 | 8 |
| Forearm Length ± SD (cm) | 23.9 ± 5.7 | 17.6 ± 5.5 | 20.7 ± 3.9 |

(N = 1), or lack of discernible motion (N = 4). This resulted in a final analysis dataset of 31 participants (DMD = 15, SMA = 7, Healthy = 9). Considering the rarity of both SMA and DMD, this sample size is considered relatively large for studies investigating these conditions.

## Curve registration and shape PCA

Let $\{\beta_i : [0, T] \to \mathbb{R}, \ i = 1, 2, \ldots, n\}$ be the set of curves representing motions for $n$ subjects. In our case, it represents the gyroscope signals of y-axis collected from the sensor on dominant wrist of participants. The gyroscope was selected because it measures angular velocity, which reduces the impact of variations in limb length. Our goal is to perform temporal alignment and phase-amplitude separation of these curves. The temporal alignment of a curve is based on a time-warping function $\gamma : [0, T] \to [0, T]$ that has the following properties. A $\gamma$ is smooth, strictly increasing (i.e., its derivative is strictly positive), and is invertible with a smooth inverse. Furthermore, $\gamma(0) = 0$ and $\gamma(T) = T$. Such functions are called *positive diffeomorphisms* or *phases* and help facilitate temporal alignments. Let the set of all time-warping functions be $\Gamma$. For a curve $\beta_i$ and a $\gamma \in \Gamma$, the composition $\beta_i(\gamma(t))$ or $(\beta_i \circ \gamma)(t)$ defines the time warping of $\beta_i$ by $\gamma$.

We begin the alignment approach using the pairwise problem. Given two curves, $\beta_1$ and $\beta_2$, we seek a time warping function $\gamma_2$ such that the peaks and valleys in $\beta_2 \circ \gamma_2$ are optimally aligned to those of $\beta_1$. Historically, one would use the optimization $\arg\min_{\gamma \in \Gamma} \|\beta_1 - \beta_2 \circ \gamma\|$ to solve the alignment problem, where $\|f\| = \sqrt{\int_0^T f(t)^2 \, dt}$ represents the classical $\mathbb{L}^2$ norm. In practice, the $\mathbb{L}^2$ of a function is approximated using a finite sum from its uniformly-sampled points, $\|f\| \approx \sqrt{\left(\frac{T}{J} \sum_{j=1}^{J} f(t_j)^2\right)}$. However, this optimization has several mathematical and computational shortcomings, and a modern approach utilizes the concept of Square-Root Velocity Functions (SRVFs). The SRVF of a curve $\beta_i$ is given by $q_i(t) \doteq \text{sign}(\dot{\beta}_i(t))\sqrt{|\dot{\beta}_i(t)|}$. If we time warp a curve $\beta_i$ into $\beta_i \circ \gamma$, then the SRVF of the new curve is given by $(q_i \circ \gamma)\sqrt{\dot{\gamma}}$. This sets up the so-called *elastic* approach to curve alignment. The optimal alignment of $\beta_2$ to $\beta_1$ is given by solving the optimization problem:

$$\gamma_2 = \arg\min_{\gamma \in \Gamma} \|q_1 - (q_2 \circ \gamma)\sqrt{\dot{\gamma}}\|^2, \tag{1}$$

where $q_1, q_2$ are SRVFs of $\beta_1, \beta_2$, respectively. This optimization is solved using the efficient Dynamic Programming Algorithm (DPA) [33]. Fig 2 illustrates this optimization where Fig 2a shows an example of arm curl $\beta_1$ and Fig 2b shows the temporal rate or warping function $\gamma_1$ of that arm curl. Fig 2c shows two misaligned curves $\beta_1, \beta_2$, and Fig 2d shows the aligned curves $\beta_1$ and $\beta_2 \circ \gamma_1^{-1}$. The minimum value in Eqn 1 results in distance between the shapes of $\beta_1$ and $\beta_2$:

$$d_a(\beta_1, \beta_2) = \inf_{\gamma_2} \|q_1 - q_2 \circ \gamma_2 \sqrt{\dot{\gamma}_2}\| \tag{2}$$

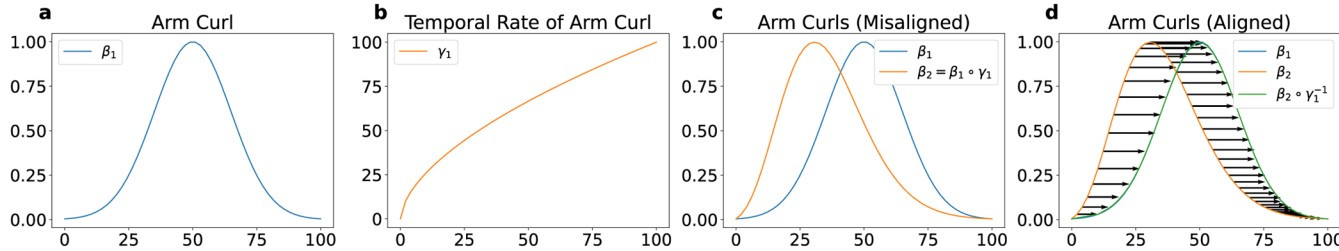

**Fig 2. A simulated illustration of the alignment of arm curls.** (a) An example of an arm curl. (b) Temporal rate or warping function of this arm curl. (c) An example of misaligned arm curls. (d) Functions after alignment.

An important property of this distance is that it is unchanged by arbitrary time warpings of $\beta_1$ and $\beta_2$. That is,

$$d_a(\beta_1, \beta_2) = d_a(\beta_1 \circ \gamma_a, \beta_2 \circ \gamma_b), \text{ for any } \gamma_a, \gamma_b \in \Gamma.$$

Therefore, it can be used to compare biomechanical signals without any influence of the rates at which the activities are performed.

This pairwise alignment can now be extended to align multiple curves and to separate their phases and amplitudes.

$$\hat{\mu}_n \doteq \arg \min_{q \in \mathbb{L}^2} \left( \sum_{i=1}^{n} \left( \min_{\gamma_i \in \Gamma} \| q - (q_i \circ \gamma_i) \sqrt{\dot{\gamma}_i} \|^2 \right) \right). \tag{3}$$

This optimization is solved iteratively. Each iteration includes two steps: (1) aligning individual SRVFs $q_i$s to the current $\hat{\mu}_n$ using Eqn. 1 repeatedly and (2) Updating the estimate of $\mu$ using cross-sectional average of current aligned SRVFs according to:

$$\hat{\mu}_n \mapsto \frac{1}{n} \sum_{i=1}^{n} (q_i \circ \gamma_i) \sqrt{\dot{\gamma}_i}.$$

We stop the iteration when the updates result in small changes. The FDASRSF [27] provides implementations of this solution in MATLAB, Python, and R. The outputs of this procedure are: (1) $\hat{\mu}_n$: the overall mean shape of the given curves, (2) $\{\gamma_i^*\}$: the phases that align individual curves to the mean shape, and (3) $\{\tilde{\beta}_i = \beta_i \circ \gamma_i^*\}$: the set of aligned curves or amplitudes of the original curves. In summary, each individual curve $\beta_i$ is decomposed into its phase $\gamma_i^*$ and amplitude $\tilde{\beta}_i$ such that $\beta_i = \tilde{\beta}_i \circ \gamma_i^*$. Fig 3 shows examples of this separation. In each row, the first column shows the original data (Figs 3a and 3e), the second column shows the phases $\{\gamma_i^*\}$ (Figs 3b and 3f), the third column shows the mean $\hat{\mu}_n$ (Figs 3c and 3g), and finally the last column shows the aligned amplitudes $\{\tilde{\beta}_i\}$ (Figs 3d and 3h). The aligned functions $\{\tilde{\beta}_i\}$ represent the shapes of given curves and can be now analyzed using Shape PCA.

Let $\{\tilde{q}_i\}$ be the SRVFs of the aligned functions $\{\tilde{\beta}_i\}$. We can calculate the covariance function of these SRVFs and obtain the principal directions of variability by performing Singular Value Decomposition (SVD) on the covariance function, $C_s = U_s \Sigma_s V_s^T$. This process is called Shape PCA because it involves conducting functional PCA in the SRVF space of the aligned functions, where the phase is already separated. After obtaining the Shape PCA principal

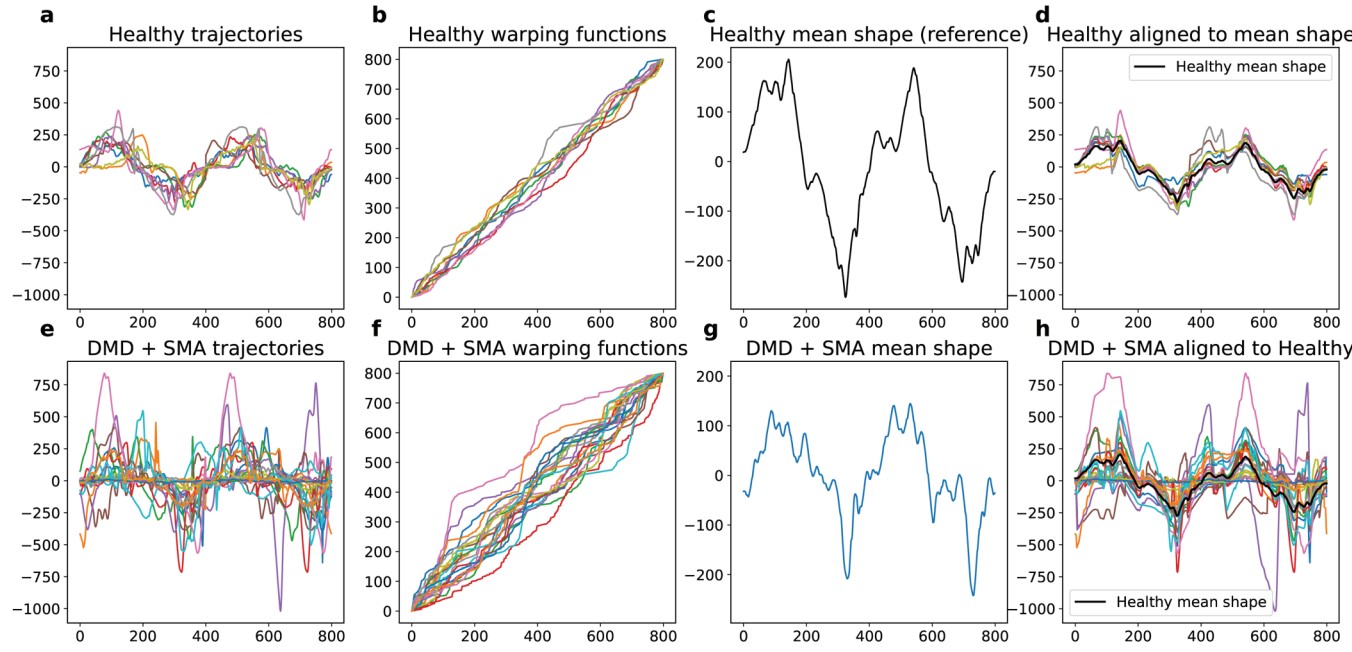

**Fig 3. (a-d) Results on performing phase amplitude separation on healthy and (e-h) DMD/SMA cohorts.**

directions, we can calculate the projections on these principal directions as $c_{s,ik} = \langle q_i, U_{s,k} \rangle$. Here, $\{c_{s,ik}\}$ represents the finite-dimensional Euclidean representations of the aligned functions or shapes and can be referred to as principal components or coefficients. These coefficients or components are also called Vertical Principal Components (VPCs).

## Statistical analysis

In order to get more robust results from Shape PCA and also handle multiple visits of participants, we run Shape PCA 100 times with a random visit taken for each subject. Then we flip the sign of SVD to get the principal components to be sign aligned with the components of the first trial. Then a mean PC score is computed across these runs as a representation embedding for each participant.

To gauge the variability in the relationship between wearable modes and clinical variables, we utilized bootstrapping. We generate a distribution of canonical correlations derived from 10000 bootstrap replicates. In each replicate, we randomly sampled participants with replacements to form a new training set (70% of the data), while the remaining 30% served as a hold-out test set. PLS was fitted on the resampled training data, and its performance, measured by canonical correlation, was assessed on the corresponding test set [34]. This approach captures the uncertainty in estimated relationships due to sampling variability. All the correlations were measured using the Pearson correlation coefficient.

For the mixed linear model regression, the random effects accounted for variation in intercepts across different participants (Participant ID), while the fixed effects included the effects of age, cohort, and their interaction. In this analysis, the $p$-values were calculated using two-sided Wald tests [35]. The significance level was set at $\alpha = 0.01$, and significance was achieved when the interaction effects were statistically different from zero, indicating a significant influence of these interactions on the dependent variable. Additionally, $p$-values were adjusted

for multiple comparisons using the Benjamini-Hochberg method [36]. Shape PCA, PLS, and mixed linear model regression were performed using the FDASRSF [27], Scikit-learn [37], and statsmodels [38] packages, respectively. All other analyses were conducted using Python 3.11.

## Results

### Insights from curve registration

To illustrate phase amplitude separation with an example, we initially generate data with a symmetric shape and purely amplitude variation (Fig 4a). To demonstrate phase variability, we generate several temporal warping functions (Fig 4b). These warping functions indicate the rate at which a motion is performed (slower or faster). Combining the amplitude variation with these warping functions results in both phase and amplitude variation (Fig 4c). The mean of these functions yields the red curve, which is asymmetric (Fig 4f), despite the original shapes being symmetric. However, performing phase amplitude separation separates the horizontal variation from the vertical one. This process temporally aligns the functions (Fig 4d), recovers the warping functions (Fig 4e), and a mean shape (depicted in blue) that is symmetric (Fig 4f). This technique provides a much more accurate representation of the original shape.

In Fig 3, we present the results of phase-amplitude separation applied to arm curl trajectories from two groups: healthy participants in the top left plot (Fig 3a) and participants with DMD/SMA in the plot below (Fig 3e). The raw trajectories, particularly from the healthy cohort, exhibit phase variability, where similar shapes occur at different times across different trajectories. Phase-amplitude separation is applied specifically to the healthy trajectories, aligning these functions temporally and deriving a mean shape. The resulting elastic mean shape of healthy arm curls is depicted in the third plot (Fig 3c), accompanied by the corresponding temporal warping functions shown in the second plot (Fig 3b). These warping functions illustrate the variability in phase alignment across different trajectories within

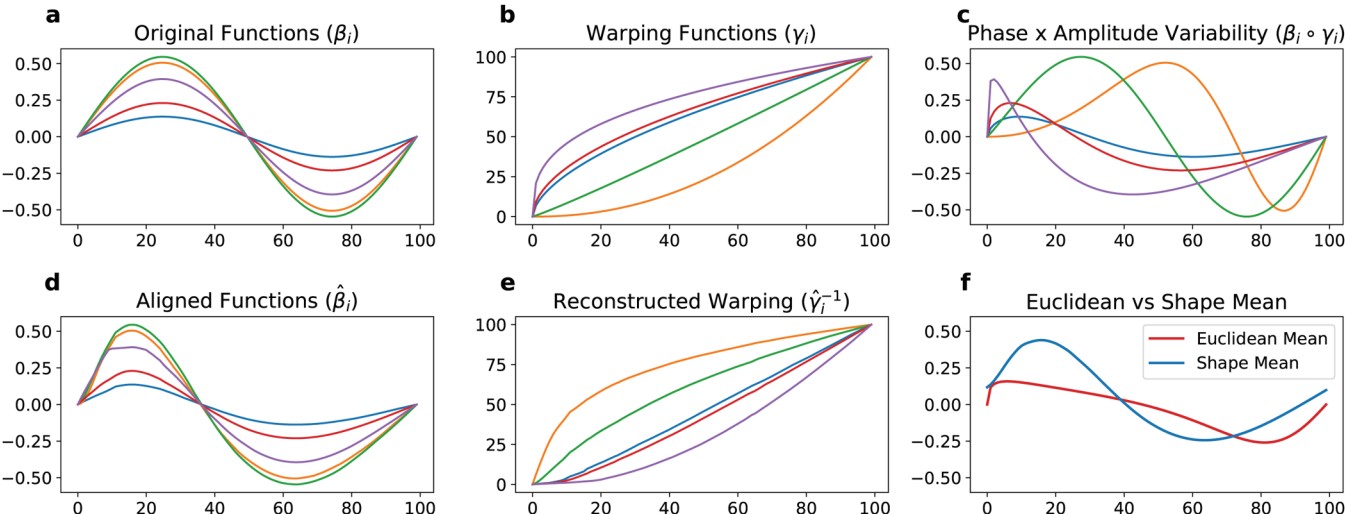

**Fig 4. Results on performing curve registration and Fréchet mean calculation with temporal matching.** (a) Signals with only amplitude variability, (b) Warping functions, (c) Signals with amplitude and phase variability, (d) Signals after registration, (e) Reconstructed warping functions, (f) Euclidean and Shape mean. Note how the shape mean (blue) captures the symmetric shape better than the Euclidean mean (red).

the healthy group. From the top right plot (Fig 3d), we observe that the peaks and valleys of the healthy trajectories align closely with the healthy mean shape, indicating effective alignment.

In the second row of Fig 3, we depict the trajectories of participants with DMD/SMA (Fig 3e). Applying phase amplitude separation within this group, we compute the mean shape of DMD/SMA, shown in Fig 3g. In Fig 3h, we align the DMD/SMA trajectories not to their own mean but to the mean shape derived from healthy participants. This approach aims to highlight deviations from the healthy mean shape. Here, we observe a notable disparity between the peaks and valleys of the DMD/SMA cohort and the healthy mean. As depicted visually in Fig 3f, the DMD/SMA trajectories require substantial warping to align them with the healthy mean, indicating greater shape variability compared to the healthy trajectories.

## Discovering modes of variation in trajectories

In Figs 5a-c, we conducted Shape PCA on arm curl trajectories across all cohorts to identify key patterns of variation. The first principal component (VPC1, Fig 5a) primarily reflects changes in angular speed while maintaining a consistent curl shape. Starting from the mean shape ($\mu$, depicted in black), moving one standard deviation along the positive direction of VPC1 ($\mu + 1\sigma v$, shown in red) reveals a reduction in angular velocity. This pattern explains 50.86% of the variance across all participants.

The second mode of variation (VPC2, Fig 5b) illustrates asymmetry in the motion. Starting from the mean shape ($\mu$, depicted in black), progressing one standard deviation along the positive direction of VPC2 ($\mu + 1\sigma v$, shown in red) reveals a decrease in the height of the peak of the curl while the trough remains unchanged. This pattern explains 27.53% of the

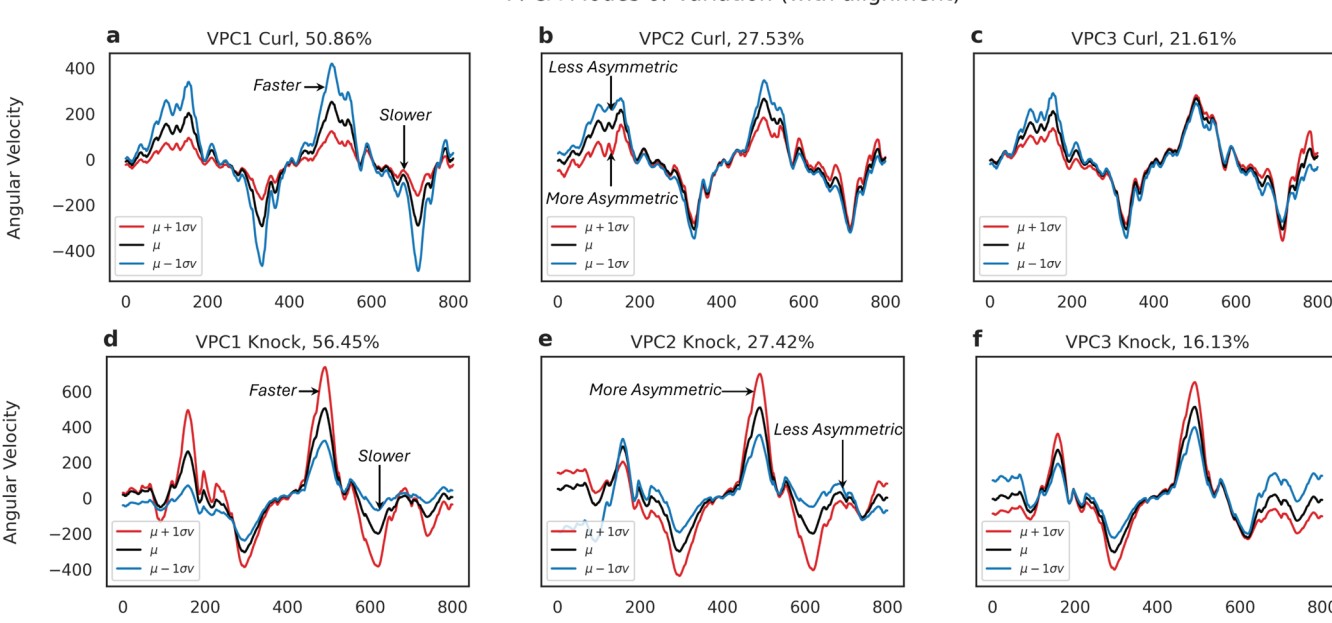

**Fig 5. (a-c) Vertical modes of variation obtained from shape PCA on the curl data.** (a) The first mode represents scaling, (b) the second asymmetry in motion while (c) the last represents noise. (d-f) Modes of variation obtained from knocking data. (d) The first mode represents scaling. (e) The second mode represents asymmetry in motion while (f) the last represents sensor noise.

variance across all participants. To validate this observation, we examine joint velocity vectors for two participants (Fig 6). This analysis indicates that these participants face difficulty during the upward motion phase, while the downward phase occurs more quickly, possibly influenced by gravitational effects. The third mode of variation (VPC3, Fig 5c) captures variability in the trajectory's tail. This mode likely reflects sensor noise or temporal segmentation noise.

The second row (Figs 5d-f) displays the results of Shape PCA applied to knocking motion curves. Similar patterns to those observed previously emerge. VPC1 appears to represent scaling (Fig 5d), indicating variations in the speed of the knocking motion. On the other hand, VPC2 seems to capture asymmetry (Fig 5e) between the speed of the first and second knocking motion. Finally, VPC3 reflects some form of sensor noise (Fig 5f). We also conducted Shape PCA on additional activities such as moving a paddle and twisting a door knob. However, these experiments yielded less interpretable results, with principal components showing less structured patterns. Consequently, we focus exclusively on two actions going forward: arm curls and knocking motion.

## Analyzing cohort differences

In Fig 7, we analyze differences in wearable features (X) and clinical measures (Y) among three cohorts. Boxplots are shown for several variables: Age, Brooke score, Average Echogenicity (indicating fat infiltration into tissue), and Normalized Elbow Torque (a normalized measure of strength across age ranges). Additionally, we present projections on the four modes of variation: VPC1 and VPC2 obtained from arm curl and knocking motions. Both DMD and SMA cohorts exhibit higher Average Echogenicity (Fig 7c) compared to Healthy, indicating greater fat infiltration into tissue. Consequently, they also show lower Normalized Elbow Torque (Fig 7d), suggesting reduced strength. In the second row (Figs 7e-h), we display boxplots of wearable features. Both DMD and SMA show large variance in VPC1 Curl (Fig 7e), with higher functioning patients on par with healthy individuals. Furthermore, SMA cohort demonstrate lower speed in knocking motion compared to Healthy (Fig 7g). Notably, VPC2 Curl activation (Fig 7f), which indicates motion asymmetry, is more pronounced in SMA compared to DMD and Healthy. This finding is intriguing given the biological differences between DMD, which involves progressive muscle fiber deterioration due to dystrophin

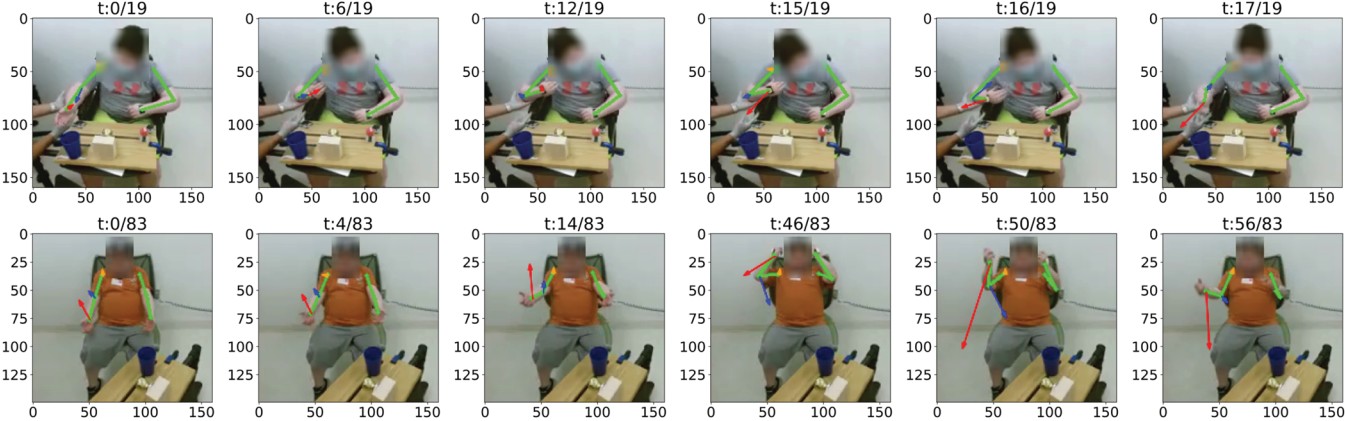

**Fig 6. Interpretation of vertical principal component 2 of arm curl (VPC2 Curl) in videos of 2 participants.** The participants performed the upward motion of the arm curl more slowly than the downward motion, likely due to the resistance posed by gravity.

## Boxplot of Clinical and Wearable Features

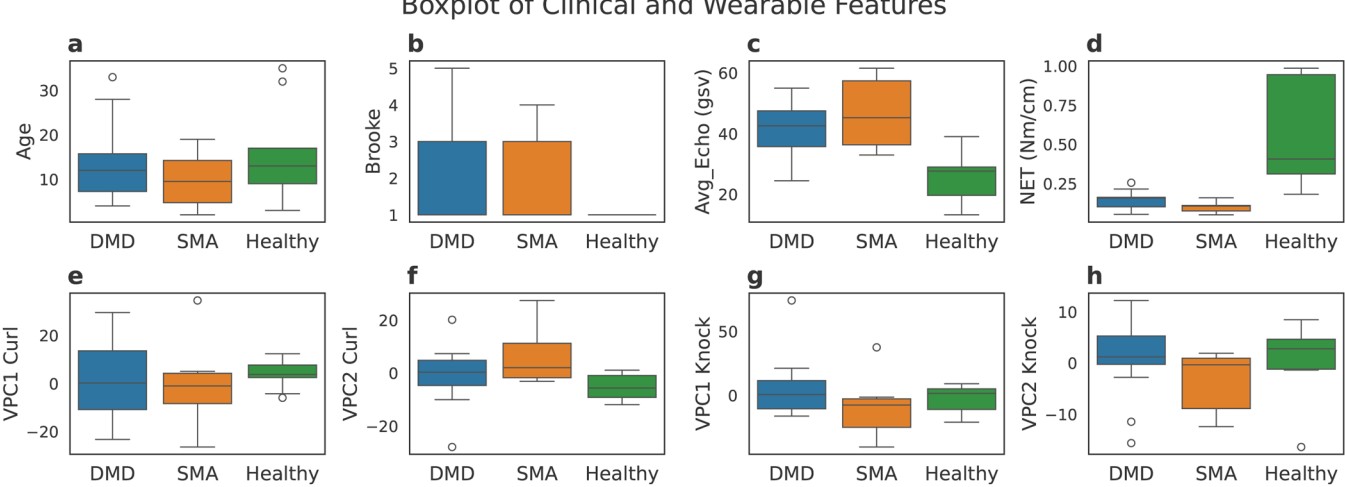

**Fig 7. Boxplots of some demographic variables along with important clinical measures and feature dimensions.** (a) Age, (b) Brooke score, (c) Average Echogenicity (Avg_Echo (gsv)), (d) Normalized Elbow Torque (NET (Nm/cm)), (e) VPC1 Curl (Speed), (f) VPC2 Curl (Asymmetry), (g) VPC1 Knock (Speed), and (h) VPC2 Knock (Asymmetry).

deficiency, and SMA, which affects spinal motor neurons. It suggests that SMA may impair subtle motion control, resulting in asymmetries in motion patterns.

## Correlations between functional modes and clinical measures

In Fig 8, we examine the correlations of modes of variation obtained from each activity with the clinical measures described in Table 1. In the top row (Fig 8a), we observe stronger correlations between VPC1 and age for DMD and SMA compared to the Healthy cohort. This positive correlation suggests that as age increases, VPC1 also increases, indicating a reduction

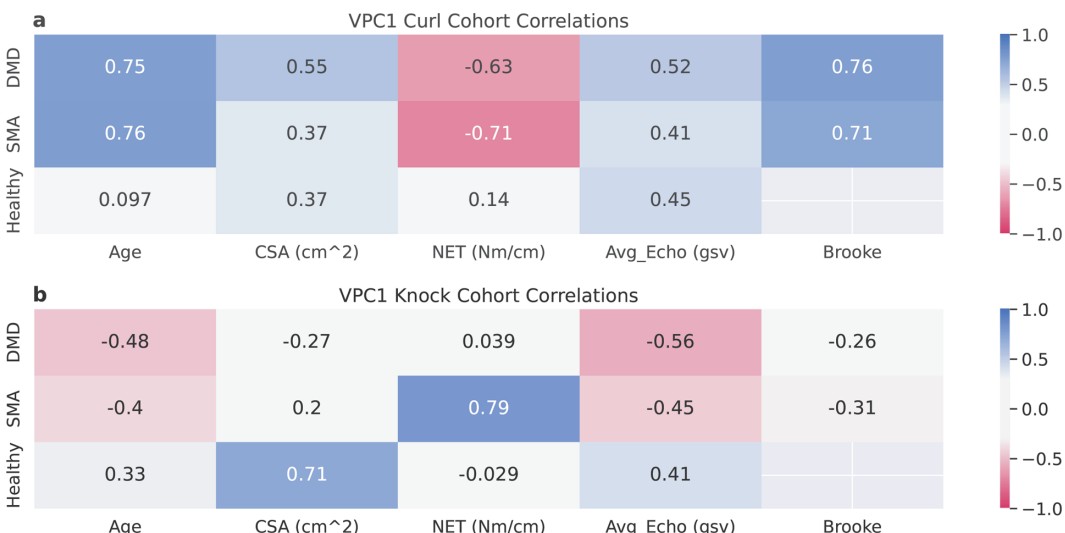

**Fig 8. Pearson cross-correlation of different VPC modes with clinical measures for DMD (N = 15), SMA (N = 7), and healthy (N = 9).** (a) Cross correlations for VPC1 Curl (speed), and (b) VPC1 knock (speed).

in angular speed. This stronger correlation in DMD and SMA may be due to the progressive nature of these diseases affecting both patient groups. An increase in VPC1 correlates with a decrease in strength, as seen in the Normalized Elbow Torque (NET). Additionally, VPC1 for DMD shows a positive correlation with Average Echogenicity (Avg_Echo), which aligns with increased fat infiltration in muscle fibers, leading to tissue weakening. In both DMD and SMA, VPC1 is positively correlated with the Brooke score, where higher scores indicate poorer muscle function. No correlation with Healthy is shown since Brooke was only collected for patient cohorts. The second row, VPC1 Knock (Fig 8b), which represents scaling in knocking motion, shows a similar but weaker correlation pattern. Since the direction of the VPC1 Knock is reversed (moving one standard deviation to the right of the mean implies an increase in speed), its correlations have opposite signs compared to the VPC1 Curl.

## Combining modes of variation

To develop a comprehensive index for assessing function in DMD and SMA cohorts (Healthy was omitted due to missing Brooke), we employed PLS to combine projections atop the principal component dimensions and correlate them with clinical variables. To gauge the variability in the relationship between wearable modes and clinical variables, we utilized bootstrapping. Fig 9 (first column) illustrates the distribution of canonical correlations derived from 10000 bootstrap replicates. As shown in the first row of Fig 9, our primary canonical dimension ($0.76 \times$ speed curl $- 0.59 \times$ speed knock $+ 0.18 \times$ asymmetry curl $+ 0.18 \times$ asymmetry knock) achieved a median canonical correlation of $r = 0.78$, with a 95% confidence interval of [0.34, 0.94] across the 10000 bootstrapped test sets. This indicates a robust association between this linear combination of wearable features and dimensions such as muscle fat infiltration (Avg_Echo), Brooke score, and age-related degenerative changes. The narrower spread of coefficients for speed of motion (VPC1 Curl and VPC1 Knock) underscores their particular significance within this dimension. Following them are asymmetry in curl motion (VPC2 Curl) and asymmetry in knocking motion (VPC2 Knock). Given the lower correlations and higher variance in coefficient estimates observed in the second and third modes ($r = 0.01$ and $r = 0.17$, respectively), we opted for the first canonical dimension as our motor function index. This decision was guided by its stronger bootstrapped correlation and more stable coefficient estimates.

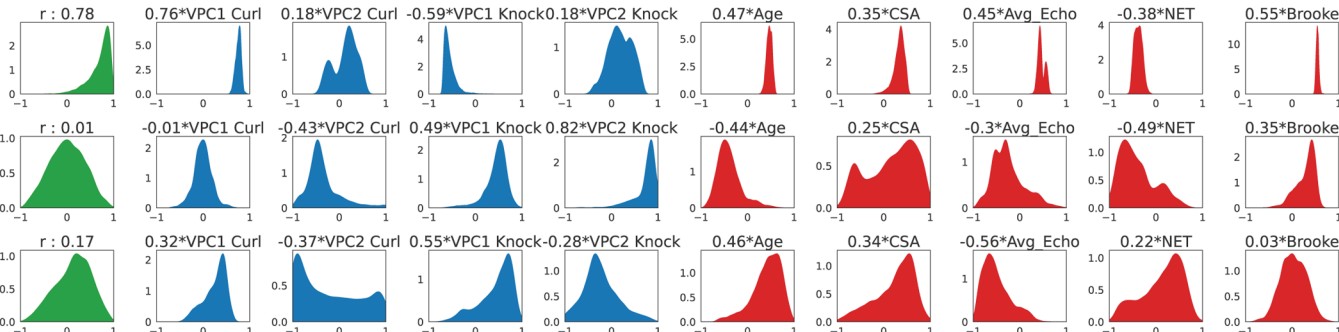

### Distribution of PLS Coefficients

**Fig 9. Distribution of canonical correlations (first column) and coefficients.** Our first canonical dimension has a median correlation of $r = 0.78$ (95% CI [0.34, 0.94]) with dimensions of muscle fat infiltration (Avg_Echo), Brooke score, and Age-related degenerative changes. Speed of curl (VPC1 Curl) and knock (VPC1 Knock) have tighter spread in distribution than the asymmetry features (VPC2 Curl and VPC2 Knock).

## Comparison with other decomposition techniques

We compared our algorithm with other low-rank decomposition techniques: specifically, Functional PCA (FPCA) without phase-amplitude separation [39] and Non-negative Matrix Factorization (NMF) [40]. The modes of variation obtained from each technique are illustrated in Fig 10, and the corresponding canonical correlations are summarized in Table 3. Our framework achieves a higher median canonical correlation and a narrower confidence interval for the first component.

## Age and VPC1 relationship

In Fig 11, we examined the relationship between age and speed of movement in DMD, SMA, and Healthy control groups. We conducted linear mixed-effects regression, modeling VPC1 Curl as an interaction between age and cohort. Specifically, for DMD ($\beta$ = 1.337, corrected $p$ = 0.001) and SMA ($\beta$ = 2.530, corrected $p$ = 0.002) cohorts, the positive slope coefficients indicate an age-related decline in the speed of curl, suggesting a loss of ability. Conversely, the

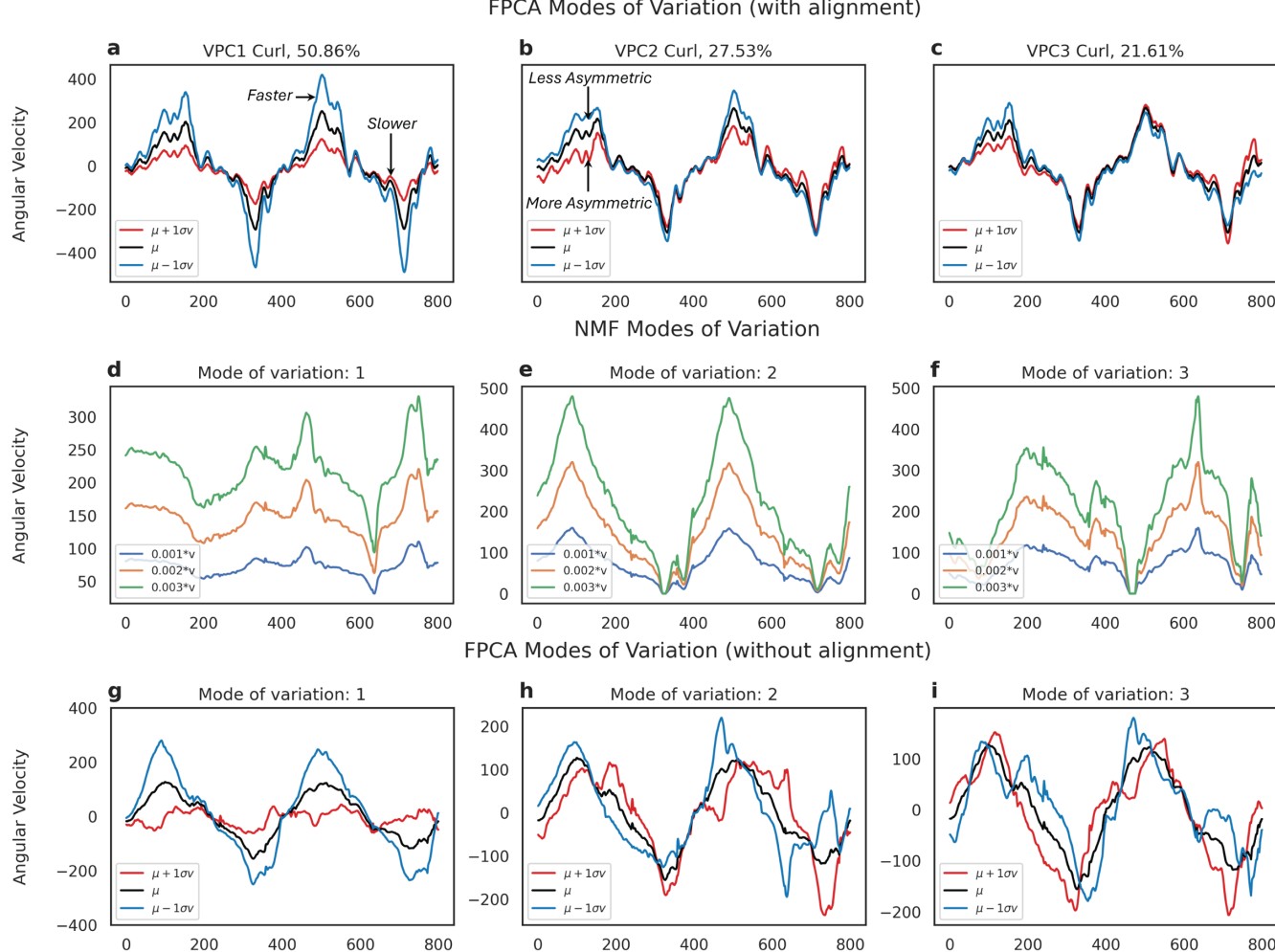

**Fig 10. Comparison of different decomposition methods, (a-c) shape PCA with alignment leads to much more interpretable modes of variation than (d-f) NMF, and (g-i) FPCA without alignment because of the phase variability.**

**Table 3. Performance comparison for different algorithms reported in terms of bootstrapped canonical correlation of each component.**

| Algorithm | Component | Median (50th percentile) | [5-95]% Confidence Percentile |
|---|---|---|---|
| **Shape PCA (Aligned)** | 1 | 0.78 | [0.34, 0.94] |
| NMF (No alignment) | 1 | 0.63 | [0.01, 0.94] |
| Functional PCA (No alignment) | 1 | 0.36 | [−0.3, 0.81] |
| **Shape PCA (Aligned)** | 2 | 0.01 | [−0.66, 0.66] |
| NMF (No alignment) | 2 | 0.28 | [−0.47, 0.81] |
| Functional PCA (No alignment) | 2 | 0.18 | [−0.60, 0.85] |
| **Shape PCA (Aligned)** | 3 | 0.17 | [−0.72, 0.71] |
| NMF (No alignment) | 3 | 0.14 | [−0.59, 0.77] |
| Functional PCA (No alignment) | 3 | −0.01 | [−0.67, 0.69] |

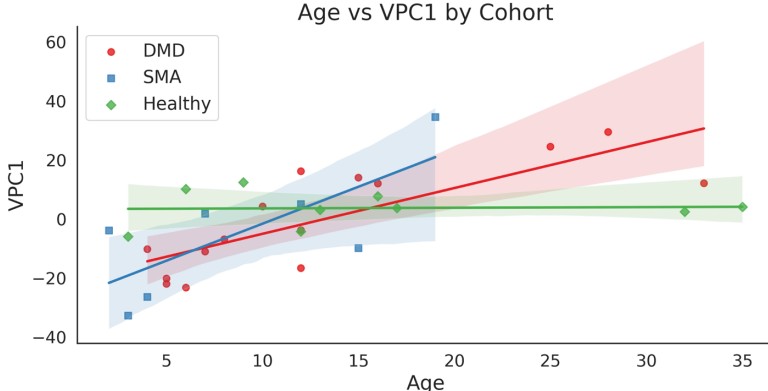

**Fig 11. Relationship between Age and VPC1 Curl in DMD, SMA, and healthy control groups.** Here, colored lines represent the regression estimated conditional mean of each cohort, and points represent the VPC1 values of each participant.

Healthy cohort did not show a significant temporal loss of function. The intercept term for individuals with DMD and SMA showed negative values, suggesting initially higher motion speeds. This finding might be attributed to the presence of higher-functioning individuals within these cohorts.

## Discussion and future work

Our approach holds promise in both clinical practice and research studies for several reasons. Firstly, by leveraging shape analysis of motion trajectories captured by wearable sensors, we extract rich, quantitative data that traditional clinical assessments may overlook. This provides a more comprehensive understanding of motor function in children with neuromuscular disorders, enabling tailored interventions and therapies. The use of Shape PCA allows us to identify nuanced patterns in movement, such as scaling and asymmetry, across various daily activities. These insights are crucial for clinicians to assess functional limitations and track changes over time more accurately than conventional methods permit.

Moreover, the PLS technique uncovers a covariation mode that correlates strongly with clinical measures like muscle fat infiltration, strength assessments, motor function indices, and age. This PLS-derived mode serves as an interpretable index of motor function, offering transparency and clinical relevance, which contrasts with the black-box nature of

many current movement analysis tools. Practically, our method supports the development of home-based monitoring systems. These systems can continuously collect data over extended periods, reducing the necessity for frequent clinic visits and enhancing patient convenience. This longitudinal data collection not only facilitates the early detection of subtle functional changes but also empowers caregivers to report on daily functions more comprehensively.

Furthermore, integrating activity recognition algorithms into these systems will enhance their utility by providing detailed insights into how children perform activities of daily living. This holistic approach paints a clearer picture of functional capabilities, aiding clinicians in making informed decisions about treatment adjustments and interventions. The non-intrusive nature of wearable sensors is particularly advantageous for monitoring disease progression, especially in patients undergoing novel therapies such as gene replacement therapy. It is also helpful for use in other pediatric populations with different neurodevelopmental problems. Telemedicine, supported by wearable sensors, enables continuous remote monitoring of participants in digital clinical trials, reducing the need for in-person visits. This approach enhances trial accessibility, supports participant retention, and ultimately improves data quality and patient outcomes.

We acknowledge several limitations of our study. While we examined multiple movements, not all yielded interpretable Shape PCA modes and were therefore excluded from the motor function index. Several factors may explain this. First, movements with high inter-individual variability may require a larger sample size to reliably estimate the covariance structure in Shape PCA. Additionally, certain tasks (e.g., rotating a doorknob or moving a paddle) produced noisier data than others, potentially due to their greater difficulty for participants compared to simpler actions like the arm curl. It is also possible that a single wearable sensor lacks the resolution to capture subtler movements, leading to challenges in segmentation and registration.

We also excluded some younger participants who were unable to follow complex instructions, which may limit the generalizability of our findings. To address these issues, we are refining our study design to focus on movements that are easier for most participants to perform-arm curl and knock, in particular, showed strong potential.

Although we used the MetaMotionR+ sensor, which is not FDA-approved, the Shape PCA approach is sensor-agnostic and compatible with consumer-grade wearables. Several FDA-approved alternatives (e.g., Actigraph LEAP [41], Empatica Embrace [42]) may be viable for clinical implementation. Finally, due to limited repeat visits, we focused on cross-sectional analysis; future work will aim to increase follow-up data to examine longitudinal changes in motor function, including progression, stability, or regression, in relation to disease course and treatment response.

Despite these limitations, we believe that our methodological approach not only advances the field of movement analysis in neuromuscular disorders but also promises practical applications in enhancing patient monitoring, clinical decision-making, and therapeutic outcomes. Future research efforts will focus on expanding participant cohorts, validating our findings across diverse populations, and refining our approach to accommodate varying clinical contexts and needs.

## Author contributions

**Conceptualization:** Shashwat Kumar, Arafat Rahman, Silvia Blemker, Rebecca Scharf, Laura E. Barnes.

**Data curation:** Shashwat Kumar, Arafat Rahman, Robert Gutierrez, Sarah Livermon, Allison N. McCrady, Laura E. Barnes.

**Formal analysis:** Shashwat Kumar, Arafat Rahman, Allison N. McCrady, Laura E. Barnes.

**Funding acquisition:** Silvia Blemker, Rebecca Scharf, Laura E. Barnes.

**Investigation:** Shashwat Kumar, Arafat Rahman, Robert Gutierrez, Allison N. McCrady, Laura E. Barnes.

**Methodology:** Shashwat Kumar, Arafat Rahman, Robert Gutierrez, Anuj Srivastava.

**Project administration:** Sarah Livermon, Allison N. McCrady, Silvia Blemker, Rebecca Scharf, Laura E. Barnes.

**Resources:** Silvia Blemker, Rebecca Scharf, Laura E. Barnes.

**Software:** Shashwat Kumar, Arafat Rahman, Anuj Srivastava.

**Supervision:** Silvia Blemker, Rebecca Scharf, Anuj Srivastava, Laura E. Barnes.

**Validation:** Allison N. McCrady, Anuj Srivastava, Laura E. Barnes.

**Visualization:** Shashwat Kumar, Arafat Rahman.

**Writing – original draft:** Shashwat Kumar, Arafat Rahman, Laura E. Barnes.

**Writing – review & editing:** Shashwat Kumar, Arafat Rahman, Robert Gutierrez, Sarah Livermon, Allison N. McCrady, Silvia Blemker, Rebecca Scharf, Anuj Srivastava, Laura E. Barnes.

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
