## [Decision Letter · Decision Letter 0]

24 Jun 2025

PONE-D-25-02163A Shape-Based Functional Index for Objective Assessment of Pediatric Motor FunctionPLOS ONE

Dear Dr. Barnes, 

Thank you for submitting your manuscript to PLOS ONE. After careful consideration, we feel that it has merit but does not fully meet PLOS ONE’s publication criteria as it currently stands. Therefore, we invite you to submit a revised version of the manuscript that addresses the points raised during the review process.

We look forward to receiving your revised manuscript.

Kind regards,

Claudia Brogna

Academic Editor

PLOS ONE

Journal Requirements:

2. Please note that PLOS ONE has specific guidelines on code sharing for submissions in which author-generated code underpins the findings in the manuscript. In these cases, we expect all author-generated code to be made available without restrictions upon publication of the work. 

Please review our guidelines at https://journals.plos.org/plosone/s/materials-and-software-sharing#loc-sharing-code and ensure that your code is shared in a way that follows best practice and facilitates reproducibility and reuse.

4. Please note that funding information should not appear in the Acknowledgments section or other areas of your manuscript. We will only publish funding information present in the Funding Statement section of the online submission form. Please remove any funding-related text from the manuscript. 

5. We note that you have indicated that there are restrictions to data sharing for this study. For studies involving human research participant data or other sensitive data, we encourage authors to share de-identified or anonymized data. However, when data cannot be publicly shared for ethical reasons, we allow authors to make their data sets available upon request. For information on unacceptable data access restrictions, please see http://journals.plos.org/plosone/s/data-availability#loc-unacceptable-data-access-restrictions.

6. In this instance it seems there may be acceptable restrictions in place that prevent the public sharing of your minimal data. However, in line with our goal of ensuring long-term data availability to all interested researchers, PLOS’ Data Policy states that authors cannot be the sole named individuals responsible for ensuring data access (http://journals.plos.org/plosone/s/data-availability#loc-acceptable-data-sharing-methods).

7. PLOS requires an ORCID iD for the corresponding author in Editorial Manager on papers submitted after December 6th, 2016. Please ensure that you have an ORCID iD and that it is validated in Editorial Manager. To do this, go to ‘Update my Information’ (in the upper left-hand corner of the main menu), and click on the Fetch/Validate link next to the ORCID field. This will take you to the ORCID site and allow you to create a new iD or authenticate a pre-existing iD in Editorial Manager.

8. Please remove your figures from within your manuscript file, leaving only the individual TIFF/EPS image files, uploaded separately. These will be automatically included in the reviewers’ PDF.

Reviewers' comments:

Reviewer's Responses to Questions

**Comments to the Author**

1. Is the manuscript technically sound, and do the data support the conclusions?

Reviewer #1: Yes

Reviewer #2: Yes

2. Has the statistical analysis been performed appropriately and rigorously? 

Reviewer #1: Yes

Reviewer #2: Yes

3. Have the authors made all data underlying the findings in their manuscript fully available?

Reviewer #1: Yes

Reviewer #2: Yes

4. Is the manuscript presented in an intelligible fashion and written in standard English?

Reviewer #1: Yes

Reviewer #2: Yes

5. Review Comments to the Author

Reviewer #1: I have carefully reviewed the manuscript. This study proposes a Shape-Based Functional Index for the objective assessment of pediatric motor function, with a particular focus on Duchenne Muscular Dystrophy (DMD) and Spinal Muscular Atrophy (SMA). Given that traditional clinical assessments, such as the Brooke score, rely on subjective measures with inherent limitations in accuracy, this research aims to develop an objective evaluation method using wearable sensors. Furthermore, as these technologies evolve, they hold promise for enabling continuous, home-based 24-hour motor function monitoring. The manuscript demonstrates a strong balance between novelty and practical applicability, making it a highly valuable contribution with significant potential for future advancements. However, one major concern is the lack of sufficient discussion on the study’s limitations.

1. Insufficient Discussion on FDA Approval of MetaMotionR+

The study employs MetaMotionR+, a wearable sensor primarily designed for research and development, which has not been approved by the U.S. Food and Drug Administration (FDA). Given that the manuscript suggests the potential clinical application of this method, the absence of discussion regarding FDA approval and regulatory considerations is a notable gap. Including the following considerations would strengthen the manuscript:

a. If this method is to be implemented as a medical device, FDA approval may be required.

b. MetaMotionR+ is currently a research-grade device, and stricter regulatory compliance would be necessary for its use as a clinical evaluation tool.

c. Future expansion of clinical trials and steps toward obtaining FDA approval should be discussed.

Without such considerations, the study remains a research-level proposal and does not sufficiently address the challenges of actual clinical implementation. If clinical application is indeed envisioned, discussion of the medical device approval process and current limitations of the technology should be incorporated.

2. Lack of Discussion on Why Certain Movements Did Not Yield Clear Results

The study analyzed five types of movements: Rotating a doorknob, Raising a cup, Arm curl, Door knocking, Moving a paddle. Among these, VPC1 Curl (arm curl speed) and VPC1 Knock (knocking motion speed) exhibited clear differences in motor function and were analyzed in detail. However, the doorknob rotation, cup lifting, and paddle movement were also subjected to Shape PCA, but their principal components were difficult to interpret, and no conclusive results were obtained. Despite this observation, the manuscript does not sufficiently discuss why these movements did not yield clear findings. Possible explanations include:

a. These movements may exhibit significant individual variation, making it difficult to derive unified principal components via PCA.

b. Variability in movement initiation and termination may have interfered with Curve Registration, affecting alignment accuracy.

c. These movements may be less influenced by muscle strength, leading to weaker differentiation among disease groups.

d. A wrist-worn sensor alone may not have been sufficient to capture these movements accurately.

e. These tasks may not adequately reflect key features of DMD and SMA, such as muscle weakness and movement asymmetry.

Since the manuscript does not address these possibilities, expanding the discussion on why only arm curl and door knocking produced interpretable results, while other movements did not, would enhance the study’s credibility.

Addressing these points would further enhance the manuscript’s clinical and scientific relevance, increasing its overall impact and robustness.

Reviewer #2: The manuscript titled “A Shape-Based Functional Index for Objective Assessment of Pediatric Motor Function” presents an innovative and technically solid approach for assessing motor function in children with neuromuscular disorders using wearable sensor data and shape-based analysis. The study is of high quality and contributes meaningfully to the field of pediatric rehabilitation and digital health.

The methodology is one of the key strengths of this work, particularly the use of Shape PCA and phase-amplitude separation to align movement trajectories. The statistical analysis, including Partial Least Squares regression and bootstrapping, is appropriately applied and strengthens the validity of the findings. The authors demonstrate strong correlations between the proposed motor function index and clinical measures such as echogenicity, normalized torque, and the Brooke score, lending further credibility to the approach. In addition, figures are generally clear and support the narrative well.

There are, however, a few areas that would benefit from further clarification or elaboration. The paper does not include any analysis of test-retest reliability or discussion of how consistent the shape-based index would be if measured over time or across different daily activities. While the current findings are compelling, the stability and reproducibility of the index should be discussed more explicitly. Another point to consider is the potential bias introduced by the exclusion of certain participants—particularly those who were unable to complete tasks due to low function or technical issues—which may limit the generalizability of the results. A brief acknowledgment of this limitation would strengthen the transparency of the study.

Although a reduced version of the dataset and the code are made available via GitHub, and this inclusion is appreciated, the full underlying data are only available upon request. In accordance with the journal’s data policy, it is recommended to provide more detailed summary-level data or anonymized datasets when possible or to more clearly justify any restrictions.

Some visual elements, such as the vertical principal component plots, could benefit from clearer labeling or interpretive annotations to guide the reader—especially those from a clinical background. Enhancing figure legends or adding simple indicators of directionality (e.g., faster/slower motion) could improve interpretability. Lastly, there are a few minor issues related to language and formatting, such as inconsistent use of "Fig." vs. "Figure" and some long or complex sentences that could be streamlined.

In conclusion, this paper is a well-structured and valuable contribution. Addressing the above suggestions would enhance the clarity, reproducibility, and practical applicability of the work and can be achieved without the need for additional experiments

6. PLOS authors have the option to publish the peer review history of their article (what does this mean?). If published, this will include your full peer review and any attached files.

Reviewer #1: **Yes: **Hideyuki Iwayama

Reviewer #2: No

---

## [Author Response · Author response to Decision Letter 1]

8 Aug 2025

We thank the editor and reviewers for the suggestions which were extremely helpful for strengthening the paper. We have included a point by point response, a revised manuscript with track changes and a clean manuscript. We hope the revision will be satisfactory.

---

## [Decision Letter · Decision Letter 1]

31 Aug 2025

A Shape-Based Functional Index for Objective Assessment of Pediatric Motor Function

PONE-D-25-02163R1

Dear Dr. Barners

We’re pleased to inform you that your manuscript has been judged scientifically suitable for publication and will be formally accepted for publication once it meets all outstanding technical requirements.

Kind regards,

Claudia Brogna

Academic Editor

PLOS ONE

Reviewer's Responses to Questions

**Comments to the Author**

1. If the authors have adequately addressed your comments raised in a previous round of review and you feel that this manuscript is now acceptable for publication, you may indicate that here to bypass the “Comments to the Author” section, enter your conflict of interest statement in the “Confidential to Editor” section, and submit your "Accept" recommendation.

Reviewer #1: All comments have been addressed

2. Is the manuscript technically sound, and do the data support the conclusions?

Reviewer #1: Yes

3. Has the statistical analysis been performed appropriately and rigorously? 

Reviewer #1: Yes

4. Have the authors made all data underlying the findings in their manuscript fully available?

Reviewer #1: Yes

5. Is the manuscript presented in an intelligible fashion and written in standard English?

Reviewer #1: Yes

6. Review Comments to the Author

Reviewer #1: Summary

This revised manuscript addresses the major concerns raised in the initial review. The authors have clarified the limitations related to the use of MetaMotionR+ by explicitly noting its lack of FDA approval and highlighting the feasibility of employing FDA-approved consumer-grade devices (e.g., Actigraph LEAP, Empatica Embrace) for clinical implementation. They have also expanded the discussion regarding why only certain activities (arm curl and knocking) yielded interpretable Shape PCA results, providing reasonable explanations related to inter-individual variability, task difficulty, and sensor limitations. These revisions significantly improve the transparency, scientific robustness, and clinical relevance of the study.

7. PLOS authors have the option to publish the peer review history of their article (what does this mean?). If published, this will include your full peer review and any attached files.

Reviewer #1: **Yes: **Hideyuki Iwayama

---

## [Editor Report · Acceptance letter]

PONE-D-25-02163R1

PLOS ONE

Dear Dr. Barnes,

I'm pleased to inform you that your manuscript has been deemed suitable for publication in PLOS ONE. Congratulations! Your manuscript is now being handed over to our production team.

Kind regards,

on behalf of

Dr. Claudia Brogna

Academic Editor

PLOS ONE